# Tumor Temperature: Friend or Foe of Virus-Based Cancer Immunotherapy

**DOI:** 10.3390/biomedicines10082024

**Published:** 2022-08-19

**Authors:** Jason P. Knapp, Julia E. Kakish, Byram W. Bridle, David J. Speicher

**Affiliations:** 1Department of Pathobiology, Ontario Veterinary College, University of Guelph, Guelph, ON N1G 2W1, Canada; 2ImmunoCeutica Inc., Cambridge, ON N0B 1B0, Canada; 3Novometrix Research Inc., Moffat, ON L0P 1J0, Canada; 4Natural Sciences and Mathematics, Redeemer University, Ancaster, ON L9K 1J4, Canada

**Keywords:** cancer, solid tumor, oncolytic virus, temperature

## Abstract

The temperature of a solid tumor is often dissimilar to baseline body temperature and, compared to healthy tissues, may be elevated, reduced, or a mix of both. The temperature of a tumor is dependent on metabolic activity and vascularization and can change due to tumor progression, treatment, or cancer type. Despite the need to function optimally within temperature-variable tumors, oncolytic viruses (OVs) are primarily tested at 37 °C in vitro. Furthermore, animal species utilized to test oncolytic viruses, such as mice, dogs, cats, and non-human primates, poorly recapitulate the temperature profile of humans. In this review, we discuss the importance of temperature as a variable for OV immunotherapy of solid tumors. Accumulating evidence supports that the temperature sensitivity of OVs lies on a spectrum, with some OVs likely hindered but others enhanced by elevated temperatures. We suggest that in vitro temperature sensitivity screening be performed for all OVs destined for the clinic to identify potential hinderances or benefits with regard to elevated temperature. Furthermore, we provide recommendations for the clinical use of temperature and OVs.

## 1. Introduction

Solid tumors generate heat due to increased vascularization and metabolic activity [1,2]. This provides many tumors with a distinctive elevated temperature compared to surrounding healthy tissues and forms the rationale for detection and diagnosis of breast cancers using thermal imaging [1,3,4]. In contrast, necrotic regions of tumors and certain tumor types, including primary glial tumors and soft tissue lipomatous tumors, are generally cooler than surrounding healthy tissue [5,6]. Tumor temperatures below 37 °C often occur as a byproduct of reduced tumor metabolic activity and/or insufficient tumor vasculature [5]. Despite being a core aspect of the cancer phenotype, cancer research has largely ignored the importance of temperature.

Oncolytic viruses (OVs) are a promising class of cancer therapeutics with the ability to preferentially target and kill tumor cells through a multimodal mechanism of action, leaving healthy tissues relatively unaffected [7]. These mechanisms include direct oncolysis via virus replication, the expression of therapeutic transgenes, the induction of antitumor immune responses via *in situ* vaccination, starvation of tumor cells via vascular shutdown, and the lethal cell–cell fusion of infected tumor cells [7,8,9,10,11]. Despite the need for OVs to perform optimally at temperatures above and below 37 °C in solid tumors, OVs are almost exclusively tested and optimized at 37 °C. Whereas the effect of elevated temperatures on OVs is relatively understudied, it appears that some OVs may be enhanced and others hindered by elevated temperatures [12,13,14,15,16,17]. This potential divergence in functionality at elevated temperatures is compounded, as OV therapy often generates fevers in patients, further increasing the temperature at which OVs may need to perform [18]. Although a similar divergence of OV functionality at lower temperatures is possible, the absence of research in this area makes speculation difficult. The differential effect of temperature on OVs has major implications with respect to optimization of OV therapies for clinical use. Furthermore, current animal models used in cancer research poorly recapitulate the temperature profile of humans; mice are unable to generate fevers, whereas dogs, cats, and non-human primates have higher baseline body temperatures than humans [19,20,21,22,23]. Overall, a significant refocusing of the cancer and OV fields on understanding the role of temperature is needed.

In this review, we describe the interplay between tumors, OVs, and temperature. A particular focus is placed on elevated temperatures and how OVs, whether heat-sensitive or heat-enhanced, can be optimized for improved anticancer efficacy, providing recommendations for researchers, clinicians, and regulatory agencies.

## 2. Tumors and Temperature

The heat-generating capacity of tumors is a dynamic variable, changing throughout their progression. Two key parameters work in concert to determine the heat-generating capacity of a tumor: tumor cell metabolism generating heat as a byproduct and vascularization within and around a tumor, which supplies the oxygen and nutrients required for tumor cell proliferation [1,2]. In the early stages of tumor development, a lack of vasculature limits tumor cell metabolism, tumor growth, and, ultimately, heat generation [24,25]. For example, Gimbrone et al. demonstrated that without access to a sufficient blood supply, the growth of a tumor is limited to 1–2 mm^3^ in diameter [25]. In this situation, a lack of sufficient oxygen causes tumor hypoxia, acting as an angiogenic switch to initiate the formation of new blood vessels [26]. This process, like the act of releasing a foot off the brake pedal, sparks exponential tumor growth, with maximal heat generated at the same time. In later stages of progression, tumors may experience a reduction in heat generation due to the development of necrosis. The longstanding belief as to the development of necrosis is that a tumor outgrows its blood supply due to an imbalance of metabolic activity and vascularization (Figure 1). In contrast, Markwell et al. argues that this late-stage tumor necrosis occurs naturally because of intravascular thrombosis, likely initiated by the overexpression of pro-coagulants by tumor cells [27]. The shutdown of tumor blood vessels then leads to local hypoxia, necrosis, and reduced tumor cell metabolic activity, which are associated with reduced tumor temperature. Subsequently, tumors may again experience localized increases in temperature as the development of hypoxia and necrosis restructure the tumor microenvironment (TME), favoring invasiveness and accelerated tumor growth [27]. As demonstrated for some glial tumors, the heterogeneity of metabolically active tumor cells and regions of necrosis can result in tumors with a marble-like thermal phenotype, with regions of higher and lower temperatures [28]. As identified by Li et al., the hypoxic area surrounding regions of necrosis in glial tumors contains brain tumor stem cells [29]. Overall, the temperature of a tumor varies during its progression and is ultimately a byproduct of vascularization and high metabolic activity.

The heat-generating characteristics of tumors have led to the use of thermal imaging for the detection and diagnosis of cancers, particularly breast cancers [1,3,4]. Elevated temperatures have been demonstrated for a handful of cancer types, including those of the breast, bladder, lung, skin, and brain [1,3,4,5,28,30,31,32]. Studies examining the temperature of lung and bladder cancers revealed an average elevated temperature of ~1 °C compared to surrounding normal tissues [30,31]. Interestingly, Yahara et al. determined the average temperature of breast tumors to be 1.79 ± 0.88 °C higher than that of the surrounding tissue [1]. A similar difference was reported by Zhao et al., who identified an average of 1.33 °C elevation in breast tumor temperatures in comparison to patient armpit temperature. This equates to an absolute tumor temperatures ranging from 37.17 to 38.44 °C [33]. The importance of vasculature in determining tumor temperature cannot be understated, as studies have found that increased blood flow and microvessel density correlated with tumor temperatures [1,30,31]. Elevated temperatures have also been detected for metastatic brain tumors [5,28,34]. For example, in a case study examining a patient with metastatic intracortical melanoma, Kateb et al. found an average temperature difference of 1.7 °C [5]. The absolute cortical temperatures recorded for the tumor and healthy brain tissues ranged from 33.5 to 36.5 °C and 33.1 to 33.5 °C, respectively. During surgery, the surface of the brain is ~4 °C cooler than the normal physiological temperature (37 °C) because of the lower operating room temperature (19–20 °C) [5]. Therefore, the actual temperature range of some metastatic brain tumors may be closer to 37.5–40.5 °C, which suggests an overlap with fever-range temperatures (38–41 °C) and hyperthermic treatment (38–45 °C) used in the clinic [35].

Although tumors tend to be elevated in temperature, certain tumors and tumor types have been associated with reduced temperatures. In a study measuring the cortical tumor temperature of six patients with brain tumors (two with metastatic tumors and four with astrocytomas), Koga et al. identified an average temperature reduction of ~2.0 °C compared to surrounding healthy tissues [36]. This corresponded to absolute temperature ranges of 31.1–35.6 °C and 33.0–36.6 °C for tumors and healthy cortex, respectively. Several studies have identified that primary brain tumors of glial origin tend to be hypothermic or lower in temperature than the surrounding healthy tissues [5,28,34]. These studies provide evidence for a distinction between hypothermic primary brain tumors and hyperthermic metastatic brain tumors. Kateb et al. suggested that factors contributing to the lower temperature of primary brain tumors may include a low density of tumor microvessels, lower metabolism in the area surrounding the tumor, greater cerebral spinal fluid in the surrounding tissue, and tumor necrosis [5]. In addition to brain tumors, soft tissue tumors, such as lipomas and atypical lipomatous tumors, have the potential for reduced temperatures [6]. In a thermographic study of soft tissue tumors, Shimatani et al. identified a slight reduction in temperature (0.05 ± 0.17 °C) of the skin located superficial to tumors in 30% (30/100) of patients. Although not a direct measure of core tumor temperature, these findings raise the possibility of reduced temperatures for soft tissue tumors. Shimatani et al. proposed that poor tumor blood flow may be one of the factors contributing to the lower temperatures they identified, based on a previous study demonstrating that lipoma tumors have lower internal vascular flow [6,37]. These studies support the role of reduced vasculature, metabolism, and the presence of tumor necrosis in determining lower tumor temperatures.

The temperature of a tumor can also be affected by therapeutic treatments, as some treatments can generate large numbers of necrotic tumor cells through direct killing or through vascular disruption, thus decreasing the overall metabolic activity and heat generation of a tumor [38]. Tepper et al. showed that following treatment, the temperature of DA3 murine mammary carcinomas was reduced and correlated with core regions of necrosis [38]. In contrast, treatments that cause patients to experience systemic fevers, such as OV-based immunotherapies, are expected to cause transient increases in tumor temperature [18]. To the best of our knowledge, no studies have reported the temperature of a patient’s tumor following OV administration. However, we postulate that the temperature of a patient’s tumor would increase proportional to the intensity of the fever they experience. This suggests that the temperature an OV faces would vary depending on the baseline temperature of the tumor and the intensity of a patient’s fever. In addition to fever, OV treatment often results in the generation of a robust antitumor immune response accompanied by an influx of highly metabolically active and proinflammatory leukocytes, potentially contributing to enhanced heat generation in a tumor [39]. Therefore, in the case of treatment by OVs, fever and the resulting inflammatory profile of a tumor may contribute to a transient increase in temperature.

## 3. Tumors, OVs, and Heat Shock Proteins

Heat shock proteins (Hsps) are highly conserved, stress-responsive proteins that are substantially upregulated in response to a wide variety of chemical and physical stressors, most notably heat shock [40,41]. Many Hsps function as molecular chaperones, helping to refold misfolded or aggregated proteins during times of cellular stress to promote cell viability [41]. Hsps further promote cell survival through the inhibition of apoptosis and by directing unsalvageable proteins for degradation via the proteasome [42,43,44,45,46,47]. Cancers are dependent on Hsps for their development, progression, and survival—a phenomenon extensively reviewed by Lang et al. and Seclì et al. [48,49]. In short, Hsp synthesis is active in a wide range of tumor cells, resulting in the overexpression of key Hsps, including Hsp27, Hsp70, and Hsp90. The significance of elevated Hsps is underpinned by their essential roles in mediating traits intrinsic to tumor cells, such as dysregulated cell division, escape from programmed cell death and senescence, de novo angiogenesis, and increased invasion and metastasis [48]. Hsps overexpressed by tumors can also function extracellularly to modify the TME for immunosuppression via binding to leukocytes, as well as transfer to other cell types via tumor-derived exosomes [49]. For these reasons, the inhibition of Hsps has emerged as a novel anticancer strategy [47]. The majority of Hsp inhibitors designed to date target Hsp90, as Hsp90-dependent substrates are associated with all ten hallmarks of cancers [50]. Unfortunately, targeting individual Hsps and using Hsp inhibitors as monotherapies have not been highly effective clinically due to feedback loops within the heat shock response, as well as issues with toxicity. For example, Hsp90 inhibition leads to a strong induction of Hsp70, thus allowing Hsp70’s cytoprotective capabilities to protect cancer cells from death [47]. Furthermore, Hsps are essential components of the stress response in healthy cells, contributing to potential drug-induced toxicity.

A large number of viruses, including most OVs, rely on Hsps as essential components for replication (Table 1) [51,52]. Interestingly, due to its essential chaperoning capabilities, Hsp90 appears to be the most widely exploited Hsp among viruses [50,51,52]. For example, Hsp90 is involved in stabilizing numerous viral structural and non-structural proteins, as well as cellular factors exploited by viruses for replication [51]. Due to the essential nature of Hsps in the replication of a broad range of viruses, Hsp inhibitors have demonstrated significant antiviral activity [51]. Although Hsp90 inhibitors have demonstrated the most potential as antivirals, it is also important to note that among Hsp inhibitors, those targeting Hsp90 are currently overrepresented in drug libraries because many Hsp90 inhibitor analogues have been developed in the cancer chemotherapeutics field [50,53].

The cancer treatment field has arrived at an understanding that monotherapies, even those using OVs, will not consistently achieve curative outcomes. As such, combination approaches with OVs have garnered significant interest. Given that both OVs and cancers rely on Hsps, the combination of Hsp inhibitors and OVs for the treatment of solid tumors seems counterintuitive. However, this combination has proven to be effective for some OVs. For example, measles virus (MeV) requires Hsp90 for the stabilization of its viral polymerase to allow for transcription of viral genes and subsequent replication [54,55]. However, the combination of MeV and the Hsp90 inhibitors, geldanamycin (GA) or its less toxic analogue, 17-allylamino-17-demethoxygeldanamycin (17-AAG), augmented MeV cytotoxicity in a few types of cancer cell lines [56]. The current understanding of the mechanism is that following Hsp90 inhibition, Hsp70 is upregulated, which is essential for MeV replication and enhanced cytotoxicity [56]. Although both Hsp90 inhibitors and oncolytic MeV are currently in clinical trials for the treatment of malignancies, no clinical studies testing the combination of the two have been conducted [57,58].

The combination of an OV with an Hsp inhibitor for enhanced oncolytic potential has also been shown for adenovirus (AdV). The combination of an AdV overexpressing the melanoma differentiation-associated gene-7 (mda-7) with GA or 17-AAG led to enhanced killing of human lung cancer cell lines [59]. Pataer et al. reported that the enhanced cytotoxicity of this combination therapy may have been associated with the inactivation of Akt (also known as protein kinase B) by GA, a protein which is often exploited by cancer cells to prevent apoptosis [59,60]. Although not investigated by Pataer et al., it is also possible that the Hsp90 inhibitor-mediated upregulation of Hsp70 may have played a role, as the overexpression of Hsp70 has been shown to enhance the oncolytic activity of AdV in human A549 lung cancer cells [61].

Beyond MeV and AdV, the combinatorial potential of Hsp inhibitors and OVs for enhanced cancer cell killing has not been investigated. The involvement of Hsp70 in the replication cycle of other OVs supports the possibility that other OVs may be enhanced through a similar Hsp70-mediated mechanism (Table 1). For example, similarly to MeV and AdV, vaccinia virus (VACV) requires both Hsp90 and Hsp70 for efficient replication [62,63,64,65,66,67]. Hsp70 is uniquely upregulated by VACV and is required for the expression of VACV early and late genes, attesting to its importance in viral replication [63,66,67]. Therefore, it is possible that the combination of VACV and Hsp90 inhibitors may result in enhanced oncolytic activity through a similar Hsp70-mediated mechanism. Additionally, tumor-cell-adapted rotaviruses with oncolytic capacity have been shown to utilize various Hsps for entry into target tumor cells, including Hsp40, Hsp60, Hsp70, constitutively expressed Hsp70 (heat shock cognate 70; Hsc70), and Hsp90 [68]. As shown by Rico et al., these tumor-cell-adapted rotaviruses bound most strongly to Hsp70 [68]. Therefore, Hsp90 inhibition could potentially enhance viral entry through the overexpression of Hsp70.

In some OVs, Hsp90, but not Hsp70, has been shown to play a role in viral replication. Therefore, it is plausible that a combination with Hsp90 inhibitors could be detrimental to the overall efficacy for these particular OVs. Overall, the combination of OVs and Hsp inhibitors is largely understudied. Combinatorial testing for enhanced tumor cell killing has only occurred with two OVs and with only two Hsp inhibitors, GA and 17-AAG, both of which target Hsp90. Future studies should determine the effects of combining inhibitors of Hsp27 and Hsp70 with MeV, AdV, and all other clinically relevant OVs in both animal models of cancers and in humans.

**Table 1 biomedicines-10-02024-t001:** Examples of heat shock protein (Hsp) involvement in oncolytic virus replication.

Virus	Heat Shock Protein	Function	References
Adenovirus	Hsp90	Potentially involved in transcription of early and late genes Interaction with E1A protein	[53,69]
Herpes simplex virus type 1	Hsp90	Interacts with VP16 for transcription of HSV-1 alpha genes Potentially involved in nuclear transport of viral capsid protein Upregulated during late viral infection	[70,71,72]
	Hsp20/Hsp27	Overexpression inhibited replication in Vero cells	[73]
Measles virus	Hsp90	Stabilization of viral polymerase (L protein)	[54,74]
	Hsp70/72	Interacts with nucleocapsid protein Enhances viral transcripts, replication, and cytopathic effects	[15,74,75]
	Hsp40	Required for interaction with Hsp70/72	[76]
	Gp96	Function unknown; upregulated during infection	[77]
	Grp78	Function unknown; upregulated during infection	[77]
Rotavirus	Hsc70/Hsp40/Hsp60/Hsp70/Hsp90	Utilized for entry	[68]
Vaccinia virus	Hsp90	Involved in release of the viral genome from the viral core Involved in the assembly of new virions Involved in maturation of the capsid, potentially through interaction with viral core protein 4a Involved in the expression of early and late genes	[62,63,64,65]
	Hsp70	Upregulated during infection Involved in the expression of early and late genes	[63,66,67]
	Hsp27	Involved in the expression of early and late genes	[63]
	Hsp105	Required for post-replication formation of nascent virions	[62]
Vesicular stomatitis virus	Hsp90	Stabilization of viral polymerase (L protein)	[52,54,78]
	Hsp60	Required for transcriptase complex and found in virus particles	[55]
	Gp96	Required for glycoprotein binding to cells for infection	[79]

Hsp = heat shock protein; E1A = early region 1A; VP16 = virion protein 16; Grp78 = 78-kDa glucose-regulated protein; Hsc70 = heat shock cognate 70.

## 4. Combination Therapies Using OVs and Hyperthermia

The heating of tumors to temperatures of 38–45 °C, known as hyperthermic treatment, has demonstrated clinical promise in combination with chemotherapy and radiation [80,81]. As reviewed by Repasky et al., hyperthermic treatment can act as a potent immunotherapeutic by enhancing tumor vascular perfusion, tumor immunogenicity, immunological functions, lymphocyte trafficking, and cytokine activity [35]. Therefore, there is considerable potential for synergy between hyperthermic treatment and oncolytic viral immunotherapy for enhanced anticancer efficacy. Studies investigating the combination of hyperthermic treatment and OVs also have the potential to provide insight into the effect of tumor- and fever-range temperatures (38–41 °C) on OVs. A summary of the research on the combination of hyperthermia and OVs is outlined below.

### 4.1. Adenoviruses

In a clinical trial treating advanced cancers, a recombinant replication-deficient AdV vector overexpressing the p53 tumor suppressor gene (rAd-p53) was delivered intratumorally weekly and followed two days later with microwave-induced hyperthermia (42–44 °C for one hour) [82]. This therapy was tested with or without radiotherapy and demonstrated a modest improvement in overall survival. The combination of AdV with hyperthermia has also been shown to enhance cellular uptake of AdV, transgene expression, cytotoxicity, and virus yield in various tumor cell types in vitro, which translated into enhanced survival of cancer-bearing mice [12,83]. Most notably, an AdV encoding for the *Escherichia coli* cytosine deaminase (CD) herpes simplex virus type 1 thymidine kinase (HSV1-tk) fusion suicide gene (CD/TK) under control of the cytomegalovirus (CMV) promoter was combined with a 4-h 41 °C heat shock at 16 h post AdV infection [83]. This strategy resulted in a 5–20-fold increase in CD-TK transgene concentrations and enhanced tumor cell killing. This effect was further enhanced when combined with radiation. AdV has also been tested in combination with gold nanorod-mediated mild hyperthermia (42 °C) for the treatment of head and neck tumor cells [12]. Jung et al. demonstrated that hyperthermic treatment applied at different times, both pre- and post-AdV treatment, could result in a significant increase in enhanced green fluorescent protein (eGFP) transgene expression. The effect of febrile-range hyperthermia (39.5 °C) on AdV cytolytic and replicative ability in both normal and transformed cells has also been investigated. Investigating two AdVs, AdV serotype 5 and an oncoselective AdV ONYX-015, Thorne et al. demonstrated that the cytolytic and replicative abilities of both AdVs were significantly hindered in non-transformed cells at 39.5 °C [84]. In transformed cells, both AdVs retained oncolytic activity in the majority of tumor cells tested at the elevated temperature. However, decreases in effective doses were observed. Oncolytic activity and replication of ONYX-015 was enhanced at 39.5 °C in a subset of tumor cells, including the human metastatic prostate cell line LNCaP [84]. Therefore, the combinatorial benefit of fever-range hyperthermia and oncolytic AdVs is likely tumor-cell-dependent. These findings also support the notion that fever-range hyperthermia has the potential to improve the therapeutic index of oncolytic AdVs.

### 4.2. Herpes Simplex Virus Type 1

Both in vitro and in vivo studies have investigated the combination of herpes simplex virus type 1 (HSV-1)-vectored oncolytic virotherapy with hyperthermia. Notably, Eisenberg et al. examined the application of a 42 °C hyperthermia pretreatment for one, two, or four hours followed by a two-hour rest for cells at 37 °C prior to treatment with HSV-1. This combination therapy showed enhanced killing in three human pancreatic cancer cell lines, with a 10–50% increase in cell killing and a twofold increase in production of HSV-1 [13]. This study provided compelling evidence for a role of Hsp72 in the mechanism of this combination therapy, as small interfering RNA-mediated knockdown of Hsp72 abrogated therapeutic efficacy, with a concomitant elimination of virus replication. The combination of intratumorally delivered HSV-1 followed by hyperthermia at 41–42 °C for 30 min has also shown efficacy in nude rat models of hepatocellular carcinoma and ovarian cancer, demonstrating the greatest reductions in tumor size relative to both monotherapies [85,86]. One limitation of this study was the use of nude rats, which lack functional T cells, which are often key for effective anticancer immunotherapies in humans.

### 4.3. Vaccinia Virus

To date, only hyperthermia pretreatment has been investigated in combination with VACV for potential enhancement of antitumor activity. Chang et al. tested the combination of a four-hour, 41 °C hyperthermia pretreatment followed by VACV-based oncolytic virotherapy in the murine colon adenocarcinoma MC-38 cell line in vitro and in vivo [14]. They demonstrated that hyperthermia pretreatment did not significantly enhance direct tumor oncolysis or replication of VACV, but instead enhanced tumor access/targeting of intravenously delivered VACV through a mechanism of increased tumor vascular permeability. This study supports the notion that hyperthermia pretreatment can enhance vascular permeability of tumors for enhanced delivery of therapeutic agents.

### 4.4. Avian Orthoavulavirus 1

Besides AdV, *avian orthoavulavirus* 1 (AOaV-1), formerly known as Newcastle disease virus, is the only other OV that has been tested in combination with hyperthermia in humans. Schirrmacher et al., at the Immunological and Oncological Center Cologne in Germany, treated two patients—one with metastatic breast cancer and one with metastatic prostate cancer—using a combination therapy of AOaV-1, hyperthermia, and autologous antitumor dendritic cell (DC) vaccination [17,87,88]. This treatment strategy was repeatedly administered over a 1–2-year period and resulted in the long-term remission of both patients. While this combination therapy appears beneficial, a definitive conclusion cannot be made without thorough in vitro, in vivo, and clinical testing in a controlled clinical trial setting.

While no other studies have investigated the combination of AOaV-1 and hyperthermia for the treatment of solid tumors, the effect of elevated temperatures on AOaV-1 as a vaccine vector for infectious diseases has been investigated. DiNapoli et al. found that the replication of AOaV-1 was enhanced at temperatures of 38–41 °C [89]. They proposed that the ability of AOaV-1 to replicate at elevated temperatures is related to the fact that birds, the host organism of AOaV-1, have higher basal body temperatures than humans, i.e., 40–41 °C [89]. These findings are promising and support the need for studies to investigate the temperature sensitivity of AOaV-1 in the context of cancers.

### 4.5. Measles Virus

A potential combinatorial effect of MeV with hyperthermia for enhanced tumor cell killing has not yet been investigated. However, Vasconcelos et al. investigated the effect of heat shock on MeV in a non-tumor cell context and showed that heat shock enhanced the replication and cytotoxicity of MeV on Vero cells, suggesting a potential benefit in combination with hyperthermia [90]. In contrast, Oglesbee et al. demonstrated that a 30-min, 41 °C hyperthermia pretreatment of non-tumor-bearing mice followed by intracranial infection with MeV resulted in enhanced viral clearance and protection compared to controls, suggesting that hyperthermia hinders MeV in vivo [77]. However, this study also suggested that the increased viral clearance was mediated through an enhanced anti-MeV immune response, which could translate to an enhanced antitumor immune response in a tumor-bearing murine model. However, no conclusions can be made about the potential for MeV-based immunotherapies in combination with hyperthermia in a tumor-bearing situation.

### 4.6. Vesicular Stomatitis Virus

Like MeV, the combination of vesicular stomatitis virus (VSV) and hyperthermia for enhanced antitumor potential has not yet been investigated. Marco and Santoro investigated the effect of heat shock on VSV replication in a non-cancer in vitro setting [16]. Using MA104 African green monkey kidney cells, Marco and Santoro demonstrated that a short-term, high-temperature heat shock (20 min at 45 °C) significantly hindered VSV replication and virus yield. These findings support the possibility that VSV and potentially other OVs may be hindered by the elevated temperatures of hyperthermia. It is important to note that 45 °C is on the higher end of hyperthermic treatments, and further research is needed to determine what effect lower hyperthermic temperatures (38–41 °C) would have on the replication of VSV and its efficacy against tumor cells.

Based on the evidence presented above, the combination of OVs, such as AdV, HSV-1, VACV, and AOaV-1, with hyperthermia is potentially an effective anticancer treatment (Table 2). However, the potential for temperature-dependent inhibition of OVs remains if this combination is not carefully designed. For example, hyperthermia in the moderate-to-high range (41–45 °C) was primarily tested in combination with OVs, largely overlooking the clinically relevant range of mild hyperthermia (38–41 °C) [35]. Thorne et al. showed that the combinatorial benefit of hyperthermia and OVs may occur in a tumor-cell-type dependent manner and could increase the safety profile or therapeutic index of some OVs [84]. Further research is needed to better understand which OVs for which types of cancer may benefit from combination with hyperthermic treatment. The evidence presented above also supports the notion that hyperthermia has the potential to enhance OV immunotherapy when delivered before, during, or following viral delivery. Further research is warranted to determine the optimal delivery time of hyperthermic treatment for each OV. For example, while hyperthermic pretreatment was shown to enhance VACV entry into tumors, it is unclear what benefits may be achieved when delivered in parallel or following VACV delivery [14]. As hyperthermia can be repeatedly administered, we recommend that researchers test hyperthermia applied before, during, and after OV treatment [91]. A strategy using multi-dosing hyperthermia may achieve additional therapeutic benefits.

## 5. Combination Therapy of OVs and Heat-Related Transgenes and Promoters

To augment OV-based cancer immunotherapies, OVs can potentially be combined with two heat-related elements: heat-related transgenes and/or the use of heat-activated promoters. To the best of our knowledge, oncolytic AdV is the only OV that has been tested in combination with these strategies.

### 5.1. Heat-Related Transgenes and OV Therapy

Although Hsps are exploited by tumor cells to facilitate the tumorigenic phenotype, they can also be exploited as immunotherapeutic enhancers [92]. As outlined by Das et al., Hsps have the ability to chaperone and present a broad repertoire of tumor antigens to DCs, resulting in the activation of both innate and adaptive antitumor immune responses [92]. Furthermore, Hsps present on the surface of tumor cells can function as targets for natural killer-cell-mediated killing. The ability of Hsps to chaperone tumor peptides in an immunogenic manner has facilitated the development of Hsp-tumor peptide vaccines. This treatment strategy involves the surgical removal of a portion of a patient’s tumor, purifying Hsp-tumor peptide complexes, and subsequently delivering these back into a patient’s tumor. Despite having shown promise in clinical settings, difficulties in tumor resection or extracting sufficient quantities of Hsp-tumor peptide complexes has limited the applicability of this therapy. Researchers have discovered that inclusion of an Hsp70 transgene in an oncolytic AdV can overcome these limitations and further enhance therapeutic efficacy [93]. This combined approach resulted in an AdV capable of overexpressing Hsp70 that can accomplish the same function of Hsp-tumor peptide presentation, in addition to providing oncolytic activity to kill tumor cells and further enhance antitumor immunity.

AdV-Hsp70 has demonstrated effective killing of a variety of tumor cells in vitro and enhanced efficacy in vivo, capable of shrinking tumors and extending survival of mice [94,95,96]. In a phase I clinical trial, the safety and efficacy of a recombinant oncolytic AdV serotype 2 overexpressing Hsp70, designated as H103, was investigated in 27 patients with advanced solid tumors [93]. Dosages ranging from 2.5 × 10^7^ to 3 × 10^12^ viral particles of H103 were tested via intratumoral injection. Fever was reported as a common adverse event, occurring in 78% (21/27) of patients. Of those who experienced a fever, ~33% (7/21) experienced grade I, ~62% (13/21) experienced grade II, and ~5% (1/21) experienced grade III. The objective response (complete response + partial response) to H103 was 11% (3/27), with transient and partial regression of distant, uninjected tumors also observed in these three patients. Overall, this trial demonstrated that intratumoral administration of H103 was safe and showed promise of antitumor activity, warranting further investigation. Wang et al. generated an E1B55kD-deleted oncolytic AdV encoding for the heat shock transcription factor 1 (HSF1) gene, demonstrating that overexpression of HSF1 enhanced AdV-mediated oncolysis of human breast cancer cells [97]. Utilizing xenograft models of human breast or colorectal cancers, they demonstrated enhanced oncolysis and replication mediated by the recombinant AdV, which was accompanied by significant tumor shrinkage. Although this strategy and these transgenes have only been tested in the context of oncolytic AdV, we predict that a beneficial effect would be demonstrated for other OVs, especially those that have been shown to benefit from the overexpression of Hsp70 during their replication, such as MeV and VACV.

### 5.2. Heat-Activated Promoters and OV Therapy

A heat-related strategy that has shown promise in enhancing the safety of OV immunotherapy is the use of heat-activated promoters, also referred to as “hyperthermia-regulated immunogene therapy”. This strategy was thoroughly reviewed by Li and Dewhirst and will only be briefly touched upon here [98]. This strategy involves an OV, either replication-competent or incompetent, encoding a therapeutic transgene under the control of a heat-activated promoter. A typical promoter used in this strategy is that which drives the expression of Hsp70B. Following administration of the OV into a tumor, heat in the form of hyperthermia is be applied in the range of 39–43 °C, activating the promoter and resulting in the production of the therapeutic transgene. This method leads to robust transgene expression, as cells may devote as much as 90% of their protein synthesis machinery to the production of Hsps during heat shock. For example, Brade et al. showed an 800-fold increase in expression of a β-galactosidase transgene under the control of the Hsp70b promotor from an adenoviral vector [99]. This strategy also allowed for reduced toxicity of an AdV vector encoding for the IL-12 gene [100]. Under the control of an Hsp promoter and in combination with hyperthermia, systemic IL-12 toxicity in a murine melanoma model was considerably reduced with minimal effect on efficacy. Overall, this OV/heat-related strategy has shown promise for oncolytic AdV but has been largely ignored otherwise. Further work is needed to test this strategy in the context of other OVs to determine whether similar benefits could be achieved.

## 6. Cold Temperatures, Cold Shock Proteins, Oncolytic Viruses, and Cancers

In addition to temperatures above 37 °C, oncolytic viruses are tasked with performing at temperatures below 37 °C in clinical settings. For example, certain types of tumors, such as primary glial tumors and soft tissue lipomatous tumors, have been shown to be hypothermic [5,6]. In glial tumors, cancer stem cells reside in the area surrounding regions of necrosis, which are associated with lower temperatures [29,38]. However, very little is known about the relationship between cold temperatures, OVs, and cancers. In this section, we discuss the involvement of environmental temperatures, cold shock proteins (Csps), and OVs in the field of cancer immunotherapy.

Statistical and epidemiological analyses suggest a positive correlation between cold environmental temperatures and cancer risk [101,102,103,104]. For example, in an analysis of 188 countries grouped by average annual temperature, Sharma et al. demonstrated that countries within the lowest average annual temperature had the highest cancer mortality rate [102]. Although there is limited research investigating the exact mechanisms of this correlation, there are some possible explanations. One proposed theory is that oxygen free radicals generated as a byproduct of the thermogenic response to a cold environment could cause oncogenic mutations, thereby increasing cancer incidence [105]. Alternatively, the relationship between increased cancer risk and colder temperatures may be explained by a third variable, Vitamin D status. Vitamin D is naturally produced by the body through exposure to ultraviolet B radiation and is associated with reduced cancer risk [102,106,107]. Individuals living in colder countries, which are often in middle- and higher-latitude regions, have reduced ultraviolet B radiation exposure and reduced vitamin D production, serving as a potential mechanism for increased cancer risk [103,106,107]. Additional possible relationships between cold temperatures, various biological functions, and cancers have been extensively outlined by Bandyopadhayaya et al. [101]. Another explanation for the link between colder temperatures and cancers, with greater supporting research, are Csps. Csps are synthesized in response to cold shock and have one or more cold-shock domains that can bind to RNA and/or DNA to regulate transcription [108,109]. Csps also serve other functions, such as regulating translation, splicing, RNA sequestration, and mRNA stability [109,110]. Since Csps have a wide range of influence on protein regulation, they have been investigated for their role in diseases, especially cancers [109].

Y-box binding protein-1 (YB-1) is a Csp upregulated in cancer cells [111]. Numerous studies have demonstrated that YB-1 is associated with breast cancers, non-small cell lung cancers, synovial sarcomas, ovarian cancers, and prostate cancers [111,112,113,114,115,116,117]. One study utilized a murine model with suppressed YB-1 expression to investigate its role in cancers [118]. In the mice lacking YB-1, angiogenesis was disrupted, and tumor growth was prevented, proving that YB-1 plays a role in cancer progression. YB-1 can also be cleaved and secreted; therefore, extracellular functions and be detectable in the serum of patients [119]. Consequently, YB-1 and other Csps, could be used as a potential marker for cancer [120,121,122,123].

Unr (upstream of N-*ras*), also known as cold shock domain-containing E1 protein, is another Csp that regulates cell differentiation and expression of several proto-oncogenes [124]. Its oncogenic properties have been reported in pancreatic cancers, melanomas, and colorectal cancers [125,126,127,128]. For example, Liu et al. identified that Unr is overexpressed in pancreatic cancers and is required for cancer cell invasiveness and cancer development [125]. Other Csps are also under investigation to understand their association with cancers that may function through different mechanisms, such as the RNA-binding protein Lin28, which contributes to cancer stem cell formation [129]. Overall, the link between Csps and cancers has been demonstrated, but further research is required to understand how their functions may work with or against cancer therapies.

The targeted inhibition of Csps has the potential to enhance anticancer therapies. For example, Zeng et al. demonstrated that the small interfering RNA-mediated knockdown of two Csps, RNA-binding motif protein 3 and cold-inducible RNA binding protein, enhanced the therapeutic response of human prostate cancer cells to chemotherapy [130]. Zeng et al. suggested that the downregulation of Csps in cancer cells mimicked the effect of heat treatment, rendering the cancer cells more susceptible to chemotherapy [130].

Despite a relationship between cold temperatures and Csps with cancers, little research has been conducted to investigate the relationship between cold temperatures and oncolytic viruses. Although tested in a non-cancer context, one study investigated the temperature sensitivity of AOaV-1 as a vaccine vector. DiNapoli et al. demonstrated that not only was AOaV-1 replication enhanced at elevated temperatures but also that AOaV-1 replication was hindered at temperatures below 37 °C (Table 2) [89]. This evidence shows that cold-sensitive OVs may exist, emphasizing the need for further research characterizing the temperature sensitivity of OVs.

**Table 2 biomedicines-10-02024-t002:** Temperature sensitivity of various oncolytic virus platforms.

Virus	Temperature Sensitivity	Supporting Information	References
Adenovirus	Heat-enhanced	Hyperthermia enhances cellular uptake, transgene expression, cytotoxicity, and virus yield in various cancer cell lines in vitro	[12,83]
		Oncolytic and replicative ability of oncoselective adenovirus (ONYX-015) maintained or enhanced in certain tumor cells under fever-range hyperthermia (39.5 °C)	[84]
		Enhanced antitumor efficacy in vivo and in humans when combined with hyperthermia	[82]
Avian orthoavulavirus 1	Heat-enhanced	Combination with hyperthermia and autologous antitumor dendritic cell vaccination resulted in the long-term remission of two patients with metastatic breast and prostate cancer	[17,87,88]
		Replication enhanced at temperatures of 38–41 °C	[89]
	Cold-sensitive	Replication hindered at temperatures below 37 °C	[89]
Herpes simplex virus type 1	Heat-enhanced	Hyperthermic pretreatment enhanced HSV-1-mediated killing of human pancreatic cancer cell lines and HSV-1 yield	[13]
		Intratumoral delivery of HSV-1 followed by hyperthermia enhanced efficacy in nude rat models of hepatocellular carcinoma and ovarian cancer	[85,86]
Measles virus	Unclear	Replication and cytotoxicity in Vero cells enhanced by heat shock	[90]
		Hyperthermic pretreatment hindered virus replication in a non-tumor-bearing intracranial model	[77]
Vaccinia virus	Heat-enhanced	Hyperthermic pretreatment enhanced tumor targeting of intravenously delivered virus	[14]
Vesicular stomatitis virus	Heat-sensitive	Replication in MA104 cells hindered by short-term, high-temperature heat shock (20 min at 45 °C)	[16]

## 7. Temperature and Animal Models

Animal models are essential for the development of viral therapeutics. However, it is important to understand the limitations of each animal model to guide the understanding and interpretation of data generated from these models, as well as the application of findings to human health. Due to limitations or a lack of accounting for these limitations when conducting research, the average rate of successful translation from animal models to clinical cancer trials is <8% [131]. In particular, the potential role of temperature among animal models has been largely neglected in the published literature on oncolytic therapeutics and cancers. In this section, we review the role of temperature and its application in preclinical and translational animal models.

### 7.1. Murine Models

Rodents represent the most commonly used animal model in biological research due to their anatomical and physiological similarity to humans, as well as their ease of handling and maintenance [132]. Although mice and humans have similar basal body temperatures (median of 37.0 °C in humans and 36.6 °C in mice), the febrile response in mice is dissimilar to that of humans [133]. The febrile response of humans and many other mammals spans a wide range, i.e., 1–4 °C above basal body temperature [134]. Contrastingly, the body temperature of mice often decreases in response to viral infection [19,135]. The published literature suggests that murine models do not properly recapitulate the phenomenon of pyrexia that commonly presents in human patients in response to viral infections. For example, administering recombinant interleukin-1β to mice at doses at or near the maximum tolerable dose only increased their average body temperature to ~38 °C [136,137]. Similarly, subcutaneous administration of near-toxic doses of turpentine, a proinflammatory chemical associated with induction of pyrexia, only raised the body temperature of mice by ~1 °C [138,139]. Thus, promising oncolytic therapies that work well in mice may be adversely affected in humans during pyrexia, as the lack of fever in mice may result in an overestimation of the potential utility of a heat-sensitive oncolytic therapy.

A potential method to circumvent the limitation of a lack of fever in mice could be to perform whole-body hyperthermia (WBH) via external heating to simulate different grades of fever. It is important to note that WBH and fever are different processes, and WBH does not fully recapitulate a natural febrile response. One main difference is that during fever, the hypothalamus-regulated body temperature set point is raised, promoting the induction of heat via thermogenesis [35]. During WBH or local hyperthermia, because the normal hypothalamic set point is not raised, the body actively works to reduce body temperature back to homeostasis. Furthermore, while hyperthermia has been shown to result in an enhancement of leukocyte functions, this is surpassed by what occurs during natural fevers. Thus, overall, mice are a poor research model for investigations wherein the natural febrile response is relevant, such as for cancers and OV immunotherapies. We argue that inferences with respect to parameters such as OV replication dynamics, tissue distribution, and safety through testing in murine models should be made cautiously until the potential for a causal role of temperature can be directly investigated and determined.

Another temperature-related issue quietly plaguing murine-related cancer research is housing temperatures. The standard housing temperatures of mouse facilities set by regulatory agencies ranges from 22 to 26 °C, most commonly set at 22 °C [140]. The thermoneutral temperature (TT) of mice, i.e., the temperature at which the basal metabolic rate is capable of maintaining body temperature, is 29–31°C [141]. Therefore, mice used for research are housed ~10 °C lower than their TT. Eng et al. demonstrated that although the body temperature of mice housed at standard temperature (ST, 22 °C) versus TT (31 °C) did not differ significantly, the plasma concentrations of norepinephrine, a neurotransmitter responsible for the activation of thermogenesis, was nearly doubled in mice housed at ST [142]. This evidence supports the notion that mice are in a constant state of mild cold shock, a form of constant stress, causing the body to utilize thermogenesis to maintain body temperature [141,142,143]. Eng et al. also showed that concentrations of Hsps of 70, 90, and 110 remained unaltered between the two groups. However, during whole-body hyperthermia, the induction of Hsps between the two groups differed. Following hyperthermia, mice housed at ST exhibited increased induction of Hsps in brain, heart, and lung tissues compared to mice housed at TT. Eng et al. concluded that researchers working on Hsps or any type of stress response in mice should consider conducting experiments at multiple housing temperatures to obtain a clearer understanding of the results. It is paramount to recognize that cancers and the TME itself produce and are acted upon by various stressors that play major causal roles in cancer development and progression, as well as response to therapy. For example, Kokolus et al. demonstrated that being in a constant state of cold shock significantly affected murine tumor models and that tumor engraftment and tumor growth was significantly reduced at TT (31 °C) [144]. Interestingly, Kokolus et al. demonstrated no differences in tumor growth between the two temperatures when using nude mice, which lack complete immune systems [144]. Their evidence suggests that the antitumor immune response in mice housed at ST (22 °C) was hindered compared to those housed at TT [144]. This result is in agreement with other studies that have shown lower housing temperatures to negatively affect murine leukocytes, such as DCs, in both tumor-bearing and non-tumor-bearing mice [145]. Overall, the use of mice that are not in a constant state of cold shock by using thermoneutral housing temperatures could considerably increase the relevance of results generated in preclinical testing.

### 7.2. Companion Animals

The body temperature of other species within the translational cancer research pipeline, such as companion animals like dogs and cats, is often overlooked. Canines and felines are valuable translational models, as they generate cancers spontaneously, have anatomic and immune systems comparable to those of humans, and are heterogenous, outbred populations [146,147]. Furthermore, companion animals with cancers, like humans, respond to OVs with fevers, which often reach high-grade levels [148,149,150]. However, unlike mice (36.6 °C), companion animals (37.5–39.5 °C) have higher resting body temperatures than humans (~37 °C) [20,21]. Therefore, if utilizing an OV sensitive to the effects of heat, its efficacy and safety profile could potentially be underestimated and overestimated, respectively, within these models. In contrast, if utilizing an OV that is amenable to or enhanced by elevated temperatures, then efficacy or toxicity may be overestimated compared to what might be observed in humans.

### 7.3. Non-Human Primates

The U.S. Food and Drug Administration, Health Canada, and other health regulatory agencies often request that safety testing of OVs be conducted in non-human primates (NHPs) prior to testing in humans. Similar to companion animals (37.5–39.5 °C), NHPs (37–39.5 °C) have higher normal body temperatures compared to humans (~37 °C) [22,23,151]. NHPs also develop pyrexia in response to OV therapy, which has implications for testing the safety of OVs that are attenuated at temperatures >37 °C [22]. For example, Pol et al. evaluated a prime multi-boost vaccination strategy with Maraba virus in tumor-free NHPs and recorded body temperatures [22]. It was noted that cynomolgus macaques had higher baseline body temperatures than humans, with all test subjects remaining between 38–40 °C throughout the course of treatment. Therefore, studies in NHPs might lead to an overestimation of maximum tolerable doses of temperature-sensitive OVs. Therefore, we recommend that regulatory agencies assigned to oversee clinical trials consider developing a policy to determine the replication potential of OVs at temperatures that are relevant to the species in which toxicity studies will be performed.

## 8. Conclusions and Future Directions

OVs show considerable promise for the treatment of solid tumors but encounter difficulties associated with performing at temperatures above and below 37 °C. In a wide range of tumors, high levels of metabolic activity and vascularization result in temperatures 1–4 °C higher than surrounding healthy tissues, a range that may be even higher when factoring in OV-induced fevers in patients [1,5,30,31,32]. In contrast, reduced metabolic activity and tumor vasculature, as well as the presence of necrotic regions in certain tumor types, can contribute to temperatures below 37 °C [5,6,38]. Despite spanning a wide temperature range, the impact of tumor temperature on OV therapeutic efficacy has largely been overlooked. This review supports the concept that cold-sensitive, heat-sensitive, and heat-enhanced OVs exist and that tumor temperature may affect OV treatment outcomes. Although still in its infancy, most research has focused on the combination of OV immunotherapy and heat-related therapies for enhanced antitumor efficacy [12,13,14,17,83,93,99]. Most notably, the combination of hyperthermia and OVs, such as AdV, HSV-1, VACV, and AOaV-1, shows potential for enhanced efficacy in the treatment of solid tumors [12,13,14,17,82,83,85,86,88]. However, a substantial amount of work in this area remains, including determining the optimal dosing and timing of hyperthermic treatment for each OV and cancer type. The reported characterization of OV temperature sensitivity must be interpreted with caution, as the majority of supporting literature is derived from studies that utilized moderate-to-high-range (41–45 °C) hyperthermia. Therefore, it is important that future studies investigate the effect of natural tumor and fever-range temperatures (38–41 °C) on OVs.

The more accurate an animal model is at recapitulating the human scenario, the more translational relevance such studies can hold. This review provides compelling evidence that temperature is an important variable that should be taken into consideration for both tumor and OV biology at all preclinical stages. However, current animal models utilized in the cancer translational research pipeline, such as mice, cats, dogs, and NHPs, do not fully recapitulate the temperature profile of humans [19,20,21,22,23,135]. These findings raise legitimate concerns regarding resource and time management, as well as implications for safety testing. For example, murine cancer models are largely used as the first stage of OV screening in vivo, yet housing temperatures cause mice to be in a constant state of cold shock, thus affecting both tumor and immunobiology [141,142,143,144,145]. We echo the recommendation of Kokolus et al. that studies using murine cancer models should be conducted at the thermoneutral temperature (~31 °C) of mice [144]. At the other end of the translational pipeline, NHPs have historically been utilized for OV safety testing prior to phase I human clinical trials. The higher basal body temperatures of NHPs, in combination with an OV-induced febrile response, could generate very high temperatures, which may inhibit heat-sensitive OVs, potentially resulting in an overestimation of their safety [22]. Therefore, we recommend that regulatory agencies request data to determine the ability of an OV to replicate at baseline temperatures, as well as at elevated temperatures associated with febrile NHPs.

As preclinical data accumulate regarding temperature sensitivity and enhancement of OVs, clinicians will be able to use this knowledge to enhance the efficacy of OV immunotherapy for their patients. For example, while working with heat sensitive OVs, clinicians may find that placing additional importance on treating fevers with antipyretics or reducing a patient’s body temperature through the use of cold-water baths may improve therapeutic outcomes. Clinicians could also consider delivering an OV at night, when an individual’s body temperature is at its lowest (~0.6 °C lower) due to the natural human diurnal rhythm [152]. In contrast, while working with heat-insensitive or heat-enhanced OVs, a focus on combination with heat-related therapies, such as hyperthermia, can be investigated for the enhancement of treatment efficacy. As suggested by Throne et al., these strategies could also include minimizing the use of antipyretics to allow a patient’s fever to act synergistically with the OV. Careful investigation and monitoring of patients would be required to avoid the risk of increased toxicity. Similarly, in the context of cold tumors, hyperthermic treatment may be used to elevate the tumor temperature to improve the functionality of cold-sensitive OVs.

## Figures and Tables

**Figure 1 biomedicines-10-02024-f001:**
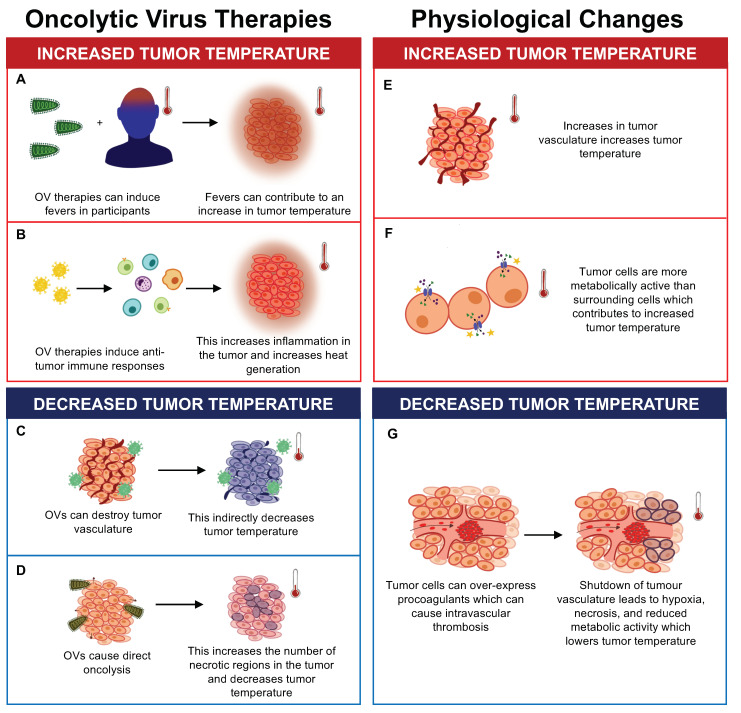
The effects of oncolytic virus (OV) therapy and physiological changes on tumor temperature. OV therapy can increase tumor temperature by (**A**) causing fevers in recipients and (**B**) inducing antitumor immune responses, which increase inflammation and heat generation. OV therapy can also decrease tumor temperature by (**C**) destroying tumor vasculature and (**D**) causing direct oncolysis of tumor cells, which increases the number of necrotic regions. Physiological changes can also impact tumor temperature. Physiological changes such as (**E**) increased tumor vasculature and (**F**) increased metabolic activity of tumor cells can contribute to increases in tumor temperature. (**G**) Tumor cells have also been shown to overexpress procoagulants which can cause intravascular thrombosis, decreasing tumor temperature due to hypoxia, necrosis, and reduced metabolic activity of tumor cells.

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
