# Peer review of "Tumor Temperature: Friend or Foe of Virus-Based Cancer Immunotherapy"

_biomedicines, 2022, doi:10.3390/biomedicines10082024_

Round 1
Reviewer 1 Report
The authors discuss the potential importance of divergent temperatures in tumors on the efficacy of treatment with oncolytic viruses. Indeed, available literature suggests that temperatures in tumors may be a bit increased or decreased compared to average core body temperature. If the temperatures in tumors are outside the range that allows replication of OVs or, alternatively, significantly promote OV replication, the topic is relevant. According to the authors, this aspect was largely overlooked in preclinical testing of oncolytic viruses. They could be right. Alternatively, divergent tumor temperatures may have been considered and found of little importance. It is therefore critical to convince readers of the relevance of testing OVs at the divergent temperatures existent in tumors.
General:
To assess how relevant tumor temperature is for treatment with OVs, one should know what the actual temperature in human tumors is. Only mentioning that it is 1-1.7 degree higher or a bit (undefined) lower than in the surrounding healthy tissue is insufficient to support the need for investigating the replication efficiency of OVs at temperatures above or under 37oC. Please provide actual tumor temperature ranges in humans. The only actual tumor temperatures presented in the manuscript now are from a single case study where this was measured on a melanoma metastasis in the brain. Here, tumor temperature was approximately 3oC higher than normal brain temperature. The actual temperature was inferred indirectly by comparison to normal body core temperature. This is a rather narrow basis to suggest, as the authors do, that actual tumor temperatures overlap with that of high-grade fevers.
As rightly pointed out by the authors, viral infections can raise body temperature and several viruses rely on induction of heat shock proteins. So, what actual temperature will be reached in an OV-infected tumor? And is the initial (uninfected) tumor temperature still relevant, or is the effect of the virus dominant? Please discuss this.
Cited observations on temperature-dependent OV functionality seem to derive mainly from artificially elevated temperatures (hyperthermia). They may therefore not be relevant for natural temperature changes in tumors, if the temperatures reached during hyperthermia treatment are outside the range of natural tumor temperatures. Please discuss.
While the existence of cold-sensitive, heat-sensitive and heat-enhanced OVs is mentioned, the presented information in this respect remains rather vague. The possible impact of a change in temperature on OV anti-tumor efficacy would be presented more clearly if the apparently known enhancement or hindrance of OV replication by low or high temperature is tabulated for the different types of viruses.
In the presentation of efficacy-promoting versus –inhibiting effects of hyperthermia I miss the distinction between transformed versus non-transformed cells. As has been shown in the past, temperature-dependent OV replication can be quite different in these two type cells, resulting in a different therapeutic index (e.g., doi: 10.1128/JVI.79.1.581-591.2005).
Specific:
Lines 229-247: It is not clear why results are presented of studies where hyperthermia was combined with replication-defective adenovirus vectors (expressing p53, GFP or prodrug-converting enzymes). These seem irrelevant for the topic. The observations do not in any way predict how elevated temperature might influence oncolytic adenovirus efficacy.
Line 359: what do the authors mean with a “type 2 AdV”? Serotype 2; second generation vector? Please define.
Lines 382-384: By definition, OVs are capable of replicating in cancer cells. It is therefore unclear why the authors consider also replication-incompetent viral vectors as OVs.
Lines 399-402: Is it true that oncolytic viruses are tasked with performing at a temperature below 37oC in necrotic tumor areas? It would seem that the virus needs to replicate in and kill viable cancer cells in viable tumor areas, rather than in necrotic areas where cancer cells are not viable anyway.
Lines 406-407: The statement that a negative correlation appears to exist between a cold environment and cancer risk is not in line with the cited observation that highest cancer mortality is seen in countries with low temperatures. The latter implies a positive correlation.
Reviewer 2 Report
This is a review article that overviews the current knowledge about tumor temperature and its impact on tumor biology and oncolytic virus therapy, and discusses clinically relevant implications on tumor temperatures towards optimizing preclinical research and clinical efficacy. This is a very well written review that carefully and extensively analyzed the existing literature and was structured with subsections with relevant and important subtopics. The issue of temperature in OV therapy has tended to be ignored in the field, and publication of this article will be valuable to many researchers in the fields of cancer and OV. I have only a few comments that might be useful to improve the clarity of this excellent work.
1. It will be very useful for the readers to create a table to summarize the temperature related characteristics of varying virus platforms (species): whether heat-sensitive, cold-sensitive, or temperature-insensitive.
2. Since lower degree hyperthermia (38-41 C) and higher hyperthermia (42-45 C) should have distinct impacts on the biology of tumors and OVs (if known), this aspect can be concisely summarized via an either figure or a table.
3. Section 3 suddenly began with Hsps in cancer. But this should be preceded by a brief description about the interaction between elevated temperature and expression of Hsps, with relevant citations, although this may be felt very basic.
4. Section 4.3. Please clarify the route of delivery of VACV, intratumoral or intravenous?, when discussing the impact of heat-related permeability of tumor vasculature.
